# A Systematic Review of the Geotechnical and Structural Behaviors of Fiber-Reinforced Polymer Composite Piles

**Fadhil Al-Darraji** [1,2,*], **Monower Sadique** [2], **Tina Marolt Čebašek** [2], **Abhijit Ganguli** [2], **Zelong Yu** [2] **and Khalid Hashim** [2]

[1] Department of Civil Engineering, College of Engineering, University of Basrah, Basrah 61004, Iraq
[2] School of Civil Engineering and Built Environment, Liverpool John Moores University, Liverpool L3 3AF, UK; k.s.hashim@ljmu.ac.uk (K.H.)
[*] Correspondence: f.k.aldarraji@2021.ljmu.ac.uk; Tel.: +44-739-1117062

**Abstract:** Composite piles have emerged as a popular alternative to conventional piling materials for deep foundations and have gained significant traction as a specific type of pile due to their potential to mitigate durability issues often associated with standard piling materials. A new type of composite piles can improve structural behavior and extend service life. This research uses an inclusive review methodology to evaluate the geotechnical and structural behaviors of fiber-reinforced polymer (FRP) composite piles. Scopus was utilized to address the relevant keywords and state-of-the-art documents, and VSOviewer software was adopted to spot recurring patterns in the data using scientometric maps. Low-stiffness composite materials are a concern, according to the research work. Thus, researchers are working on confined concrete-filled FRP piles to improve the structural and geotechnical properties used in various load-bearing conditions. However, more research is required to comprehensively understand the behaviors of the studied types of composite piles. Indeed, there is a need for large-scale lab and field studies to determine how axial and lateral loads influence composite piles. This could help create guidelines for constructing the reviewed types of composite piles.

**Keywords:** composite piles; systematic review; standard piling materials; VOSviewer software



## 1. Introduction

Pile foundations are essential for supporting superstructures such as bridges, piers, and offshore platforms. However, damages to piles are unavoidable because piles are subjected to overloading or over-extended periods in complicated environments. Piling with steel, concrete, and timber is widely employed today. Using these materials in harsh soil marine settings can lead to several problems, some of which include the deterioration of timber, steel corrosion degradation, and marine borer attack on concrete. Conventional piling materials subject to severe exposure can lead to short service life and substantial maintenance expenses [1,2]. Figure 1 shows examples of conventional pile structures that have deteriorated due to corrosion. The oldest and most common methods for protecting piles from deterioration and corrosion involve using treated timber or spraying steel with a thick conductive layer [3]. However, these pile protection strategies are expensive and negatively impact the marine environment. Because of these problems, researchers around the globe have been searching for long-lasting and efficient replacements that can withstand harsh environments. In recent years, composite piles have begun to replace traditional piles in deep foundations due to their many advantages, including stronger corrosion resistance and an improved strength-to-weight ratio [4–6].

Researchers have developed a novel structural system using a composite tube filled with concrete [7]. While the concrete contributes to an increase in the system's overall stiffness, the tube is molded to encase the material of the whole structural element. In addition, other authors have demonstrated that the suggested system performs better than

similar prestressed and reinforced concrete structural elements when subjected to combined axial and lateral loads [8]. The lack of knowledge on the history of composite piles as geotechnical concerns is one of the main concerns that prevents their widespread usage in the industries of the United Kingdom. Few references can be accessed; this knowledge deficiency is a primary challenge [4,9].

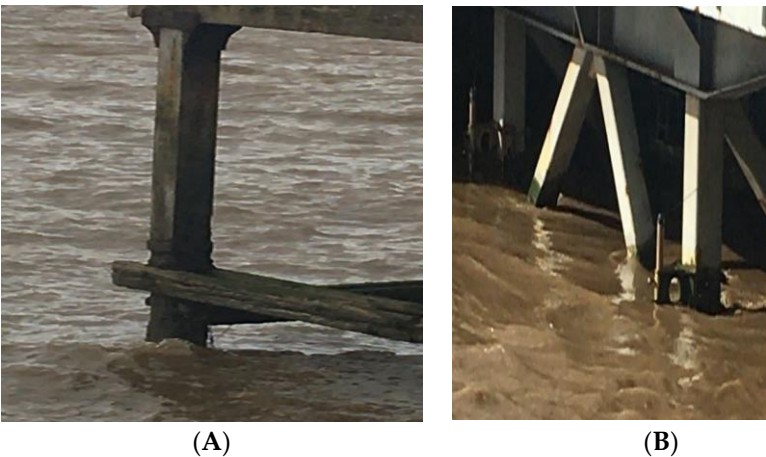

**(A)**                                    **(B)**

**Figure 1.** Traditional piles have several issues when placed in marine and other corrosive environments. (**A**) Degradation of concrete piles, and (**B**) corrosion of steel piles.

To improve the performance of composite piles, it is crucial to study the research that has already been done on the subject. A wide range of methodologies can be used to study the research concerned with composite pile development. Research metrics and data mining are leading methods for understanding and exploring research conducted about a subject. They are becoming increasingly popular techniques to assess articles already written about the subject and estimate their value based on the times the work has been referenced in other academic publications. This practice is known as bibliometric analysis. Scientometric evaluation is used in mapping a particular domain of knowledge and assesses the degree to which an article or body of research meets its objectives [10]. The primary purpose of the scientometric analysis is to evaluate the influence of nations, publications, authors, and organizations, and understand how each factor relates to the others. Combined with specialized software, these datasets can be shown in a way that demonstrates how they are connected.

To improve composite piles and narrow the gap between research and large-scale industrial implementation, this paper aims to better understand their capabilities and primary research characteristics that affect their behavior. A scientometric approach and an in-depth discussion have been utilized.

## 2. Data and Sources

This study provides a review of the previous research in the field of composite piles foundation. It was carried out in two stages: first, a quantitative stage, during which the data was retrieved from a bibliometric database, and second, a qualitative step, during which the articles were filtered and grouped manually using VOSviewer software.

A mixed technique was utilized to conduct the literature review, and bibliometric analysis combining multiple research paradigms makes it possible to investigate a topic from multiple angles, discover new insights, and more easily formulate and test hypotheses [11]. Quantitative and qualitative research approaches and the figures and explanations that justify them are combined in mixed methods research [12]. Since sustainability is a significant concern across disciplines and fields of study, this approach explored the deep foundations of composite piles.

Before choosing which database to get the information from for a systematic review, the dataset needs to be reliable and comprehensive. Scopus and Web of Science are the

two databases that include the most exhaustive and unbiased collections of scholarly work [13]. When "composite piles" was typed into the search bar on Scopus, it returned 474 results, more than the Web of Science core collection 432. Scopus is a citation and abstract database created by Elsevier. The larger dataset provided by Scopus makes it simpler to do scientometric analysis and map the findings. No time frame or language was used to search abstracts, keywords, and titles for the term "composite piles," although this yielded some interesting findings. Subsequently, the findings were further filtered by excluding articles with less than two citations. Besides, the selected papers were further refined by only considering papers written in English. The titles and abstracts of these papers were also skimmed. Articles whose content was directly related to the topic and the field of composite piles and deep foundations in construction were selected, and all other articles were excluded. Following the completion of this step, there were a total of 252 papers still in the composite pile.

## 3. Scientometric Review Interests

### 3.1. Documents Were Published According to the Year

Figure 2 shows the distribution of papers about composite piles over time. Since 1986, when the number of studies published was one per year, the number of studies on composite piles has grown noticeably. Recycled plastic covered inside a steel pipe was the first type to find use in marine construction. Los Angeles port, in 1987, saw the first implementation of this design as a panel pile in the United States [14]. After 2012, however, there was a noticeable increase in research conducted on composite piles. It is a rising trend, particularly in recent years, despite the fluctuations in certain years. The growth may be attributed to the inventory of barriers that impede progress in the composite pile, such as the long-term behavior of the material in the composite.

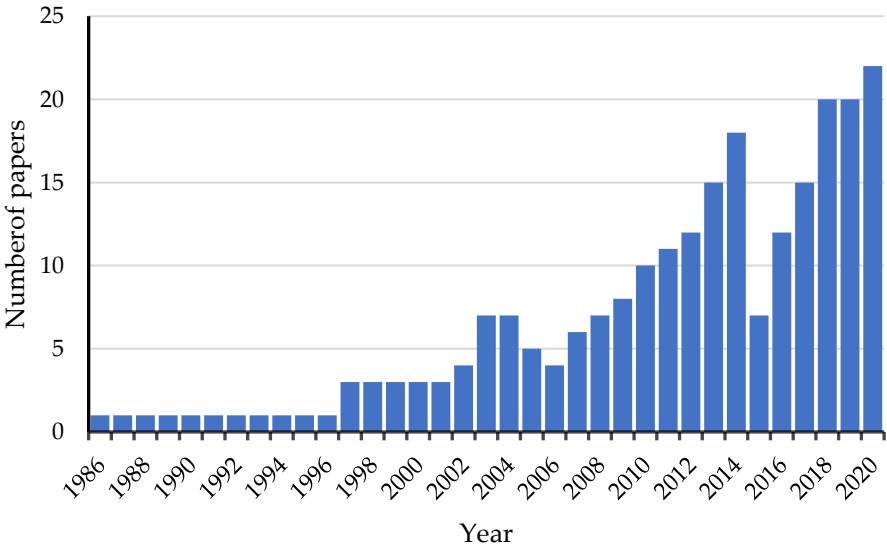

**Figure 2.** The distribution of composite piles-related publications over time.

### 3.2. Type of Document and Topic Area

Figure 3 illustrates the different types of publications received from Scopus for the study. Most documents were article and conference papers, with corresponding percentages of 50.6% and 44.7%, respectively. Figure 4 shows the distribution of the subject fields for which the documents were written, with engineering corresponding to 52.6%, materials science to 21.1%, and earth plants to 13.5% of all papers. Figure 4 also discusses the three primary subfields of composite pile research and identifies the one with the most publications.

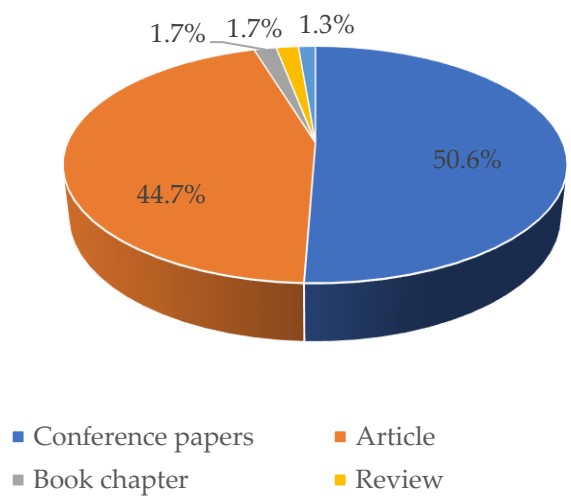

**Figure 3.** The academic publications included in this study.

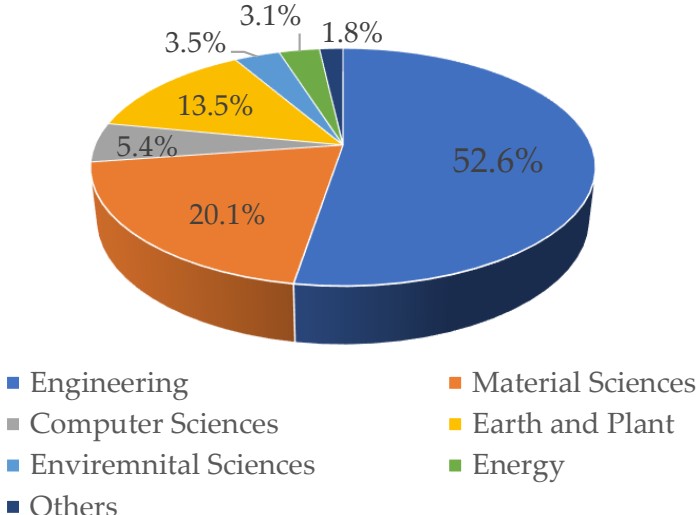

**Figure 4.** The main respective topic areas focused on the composite pile.

### 3.3. The Most Frequently Used Keywords

The keywords highlight the important ideas discussed in the articles and the main areas of study that fall within the scope of the topic [15]. Further, the occurrences of the keywords were analyzed. Only keywords that occurred at least 14 times were considered for inclusion. Connections between frequently used keywords are shown graphically in Figure 5. Total link strength (TLS) increases proportionally with the links connecting two keywords. The length of space separating keywords represents how closely related the two knowledge areas are to one another [16]. This enabled us to understand where the term composite piles are used in scientific research. For instance, composite foundations are frequently used around finite element, bearing capacity and numerical simulations. However, it can be seen that composite piles are much more used around soft soil, pile foundation, and piles. In terms of the keywords, this distance quantifies the frequency with which they are found together. The research domain and themes that are special to the study are reflected in the keywords [15].

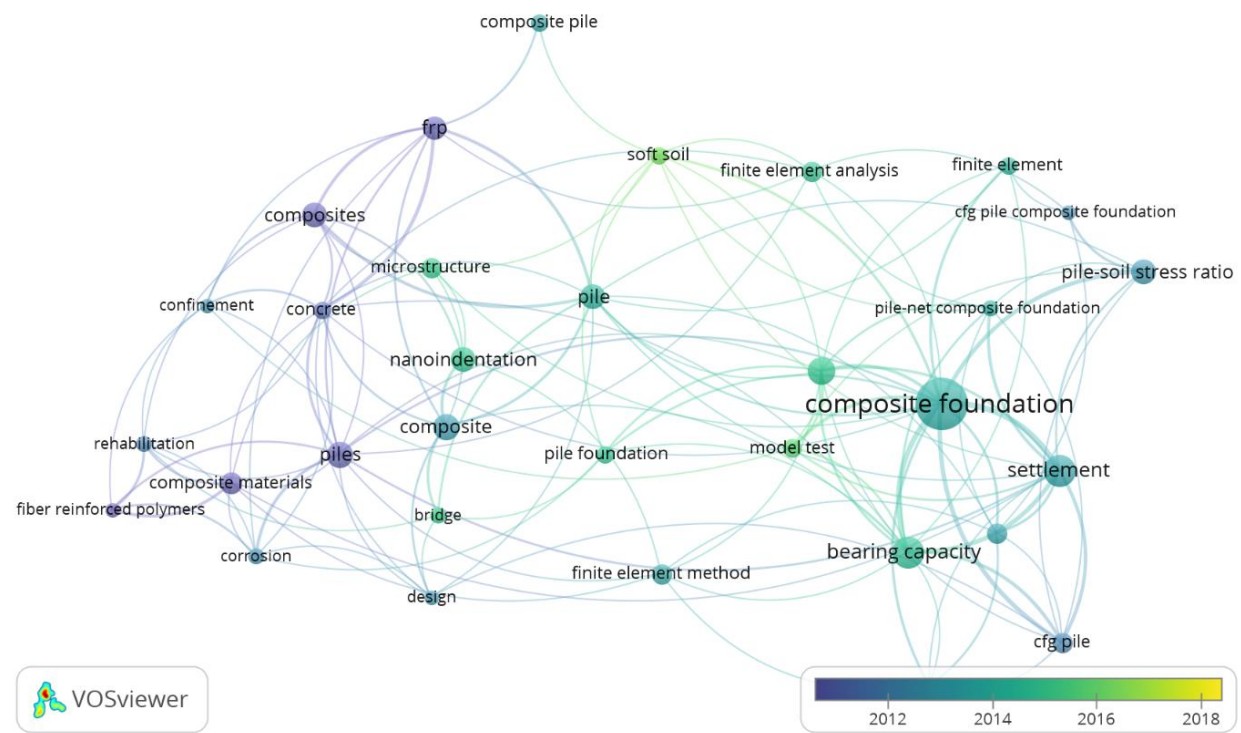

**Figure 5.** Visualization of frequently used keywords.

The keywords that most frequently occurred in the investigated papers are listed in Table 1. VOSviewer software, a bibliometric mining tool, was utilized to acquire the TLS and the occurrences. Only search terms that occurred a minimum of 20 times were considered. Geotechnical and foundational research emerge as the most prominent when comparing the keyword visualization to the table of the most often occurring phrases and filtering out the keywords which do not reflect an important study subject.

**Table 1.** Total link strength in composite piles and the most often occurring author keywords.

| S/N | Keyword | Occurrences | Total Link Strength |
|---|---|---|---|
| 1 | Composite foundation | 128 | 84 |
| 2 | Settlement | 48 | 57 |
| 3 | Bearing capacity | 48 | 40 |
| 4 | Numerical simulation | 37 | 28 |
| 5 | piles | 33 | 29 |
| 6 | Composite | 33 | 24 |
| 7 | Pile | 30 | 32 |
| 8 | Pile-soil stress ratio | 28 | 22 |
| 9 | Nanoindentation | 28 | 10 |
| 10 | frp | 25 | 29 |
| 11 | Composite pile | 25 | 31 |
| 12 | Composite material | 23 | 18 |
| 13 | Mechanical properties | 22 | 17 |
| 14 | Numerical analysis | 21 | 10 |
| 15 | Microstructure | 21 | 18 |
| 16 | Cfg pile | 20 | 25 |
| 17 | soft soil | 20 | 12 |
| 18 | Fiber-reinforced polymers | 20 | 16 |

*3.4. Country-Wise Origin of Literature*

Figure 6 is a presentation of a network analysis of nations that are actively engaged in this topic. The node size represents the amount of work done by each country. The lines

connecting the nodes indicate the degree to which the countries are interconnected. The top 10 countries in terms of the quantity of research published are as follows: China, the United States, Canada, Australia, Japan, South Korea, Russian Federation, and Turkey. This conclusion is in line with prior research that demonstrates these countries are in the lead in developing composite piles and deep foundations [17].

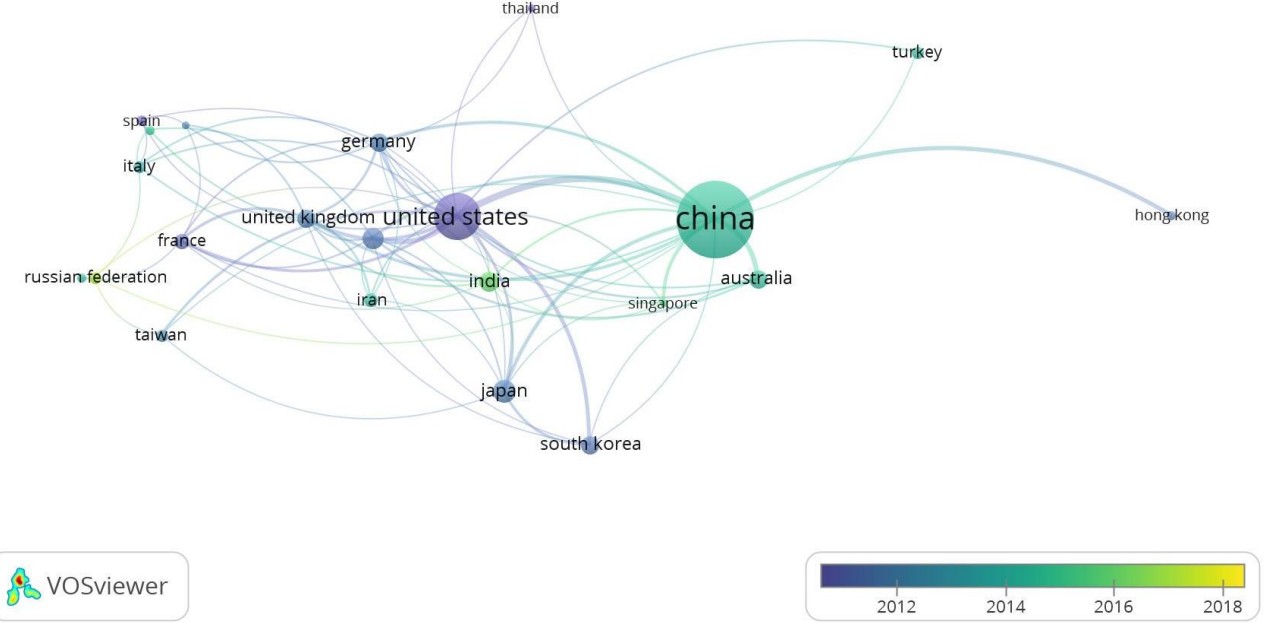

**Figure 6.** A visual representation of the cited countries.

### 3.5. Most Cited Institution

It was discovered that 805 different institutes had researched the performance of composite piles. Table 2 ranks the most influential research institutions according to their impact on the investigation. When ranking universities by how often they have been referenced, Tongji University in China, the University of New Mexico in the United States, and Chang'an University in China come out on top. The VOSviewer citation analysis is used to figure out the TLS. This takes into account how often one institution links to other institutions.

**Table 2.** Most prominent academic institutions based on how often their publications.

| S/N | Institution | Citations | TLS |
|-----|-------------|-----------|-----|
| 1 | Shanghai University, China | 114 | 10 |
| 2 | Tongji University, China | 382 | 28 |
| 3 | University of New Mexico, United States | 252 | 12 |
| 4 | Southwest Jiao Tong University, China | 97 | 19 |
| 5 | Chang'an University, China | 149 | 18 |
| 6 | Lanzhou University of Technology, China | 81 | 6 |
| 7 | Univ. of Illinois, United States | 90 | 22 |
| 8 | Hohai University, China | 84 | 9 |
| 9 | Istanbul technical university, Turkey | 79 | 5 |

### 3.6. Top-Ranked Authors

The program VOSviewer was utilized to carry out the author's co-citation analysis. It helps determine and analyze the research area's progress by showing the connections between writers mentioned in the same publication. The most frequently mentioned writers are included in Table 3, along with the TLS of each author. Han, J (203), Mirmiran (189), and Armstrong (176) are among the authors who have been cited most frequently.



**Table 3.** Top authors and their Total Link Strengths are listed.

| S/N | Author | Citations | TLS |
|---|---|---|---|
| 1 | Han, j. | 203 | 977 |
| 2 | li, j. | 117 | 830 |
| 3 | Mirmiran, a. | 189 | 514 |
| 4 | Armstrong, r.w | 176 | 366 |
| 5 | Poulos, h.g. | 126 | 333 |
| 6 | Randolph, m.f | 149 | 469 |
| 7 | Shahawy, m. | 102 | 507 |
| 8 | Wang, j. | 165 | 976 |
| 9 | Zhang, y. | 140 | 755 |
| 10 | Liu, h.l. | 124 | 261 |

As assessed by co-citation analysis, the relationships between the most influential authors in the area are depicted in Figure 7. The co-citation frequency is represented by the thickness of the line connecting each researcher, and the node's size indicates the frequency with which researchers are cited in conjunction [18].

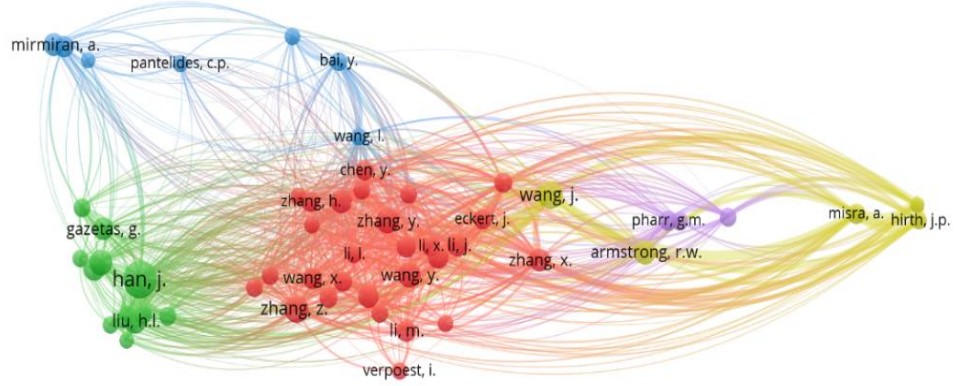

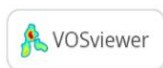

**Figure 7.** Author co-citations and their connections.

## 4. Discussion

### 4.1. Composite Pile Types

This section describes the appearance and sequence of composite pile types. Most authors' keywords connect the composite pile type with the highest TLS. The main type utilized in maritime construction comprised recycled plastic encased in a steel pipe. Then, polymer matrix composites (PMCs) were suggested for employment outside reinforcement of emergency piles [19]. This led to the development of the second type of composite piling, which is applied in maritime structural applications: FRP piles [20–22]. Due to the success of composite piles, fiber-reinforced polymer (FRP) piles were considered for use as fender piles for replacing wooden piles in numerous projects in the United States [9,14].

Five composite piles are considered viable for load bearing or fendering purposes. These are known as fiber-reinforced polymer (FRP) piles, steel core plastic (SCP) piles, structurally-reinforced plastic (SRP) piles, plastic lumber (PL) piles, and fiberglass pultruded (FP) piles [7,23]. The five varieties of composite piles that are now in use can be seen as illustrated in Figure 8.

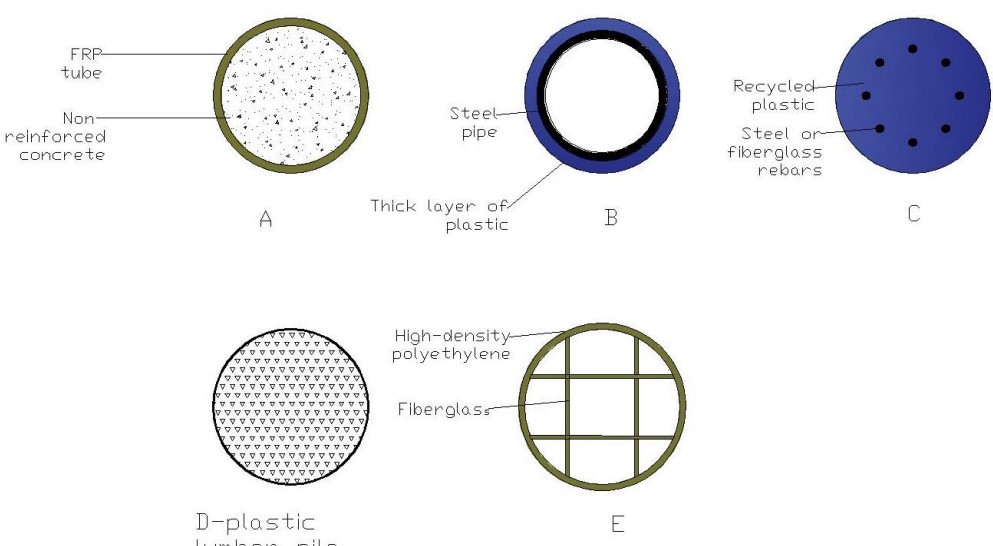

**Figure 8.** Composite pile types: (**A**) fiber-reinforced polymer (FRP) piles, (**B**) steel core plastic (SCP) piles, (**C**) structurally-reinforced plastic (SRP) piles, (**D**) plastic lumber (PL) piles, and (**E**) fiberglass ultruded (FP) piles.

### 4.1.1. FRP Piles

Acrylic-coated FRP tube sections are typically filled with non-reinforced concrete to create FRP piles. This composite pile has received great attention from researchers and has been adopted frequently [24]. The primary function of the FRP tube is to serve as a noncorroding reinforcement, to confine the concrete in compression, shield the concrete from harsh environmental conditions, and keep the concrete in place during construction. On the other hand, the concrete infill provides internal compression resistance, enhances pile stiffness, and prevents FRP piles from buckling [25].

### 4.1.2. SCP Piles

SCP piles are made of a steel pipe with a thin wall wrapped in a thick layer of plastics. The shell protects against corrosion, while the steel tubular core is responsible for the bulk load-carrying ability. There are a few concerns regarding SCP piles' structural performance in marine conditions, namely concerning the lamination between the core and the shell. One year after the installation date, cracks in the plastic shell revealed that the shell was formed using plastics unsuitable for the purpose [1].

### 4.1.3. SRP Piles

The extruded recycled plastic grid is the primary component of reinforced plastic piles, further strengthened by steel or fiberglass rebars. The composite pile develops into a recyclable material when fiberglass-reinforced plastic (FRP) rods enhance the plastic core [26]. The experimental SRP piles used in the tests displayed a large amount of deflection under lateral loading and distinctly distorted them throughout the installation [2,14].

### 4.1.4. PL Piles

PL piles are made of a matrix consisting of recycled plastic and fiberglass reinforcement randomly dispersed throughout the matrix. To reduce the overall weight, the thick hard outer tube is connected to the outer surface of the inside plastic core, which is filled with foam. According to the findings of the studies, Young's Modulus (E) of this pile type is 0.37 GPa, which is more than 40 times less than the E of concrete. As a result, most load-bearing applications cannot use this type of pile [27].

4.1.5. FP Piles

FP piles are another variety of structurally-reinforced FRP piles. These piles are made of high-density polyethene (HDPE) casing and fiberglass grid to give structural strength and limit distortion. The grid is constructed out of four plates that cross with one another, and the spaces between the plates are filled with either HDPE, plastic lumber, or polyethene foam fills. [5]. Because of their capacity to absorb the shock of passing vessels, they are only used in marine environments as sheet pile walls and fender piles. According to the findings of Lampo et al. [1], because of its poor performance under axial and lateral loads, this type of pile should not be employed in load-bearing applications. These applications are incompatible with this pile type.

*4.2. Design Composite Piles*

Bearing capacity, settlement, numerical simulation, and numerical analysis are all associated with the design of composite piles, which led to their designation as research clusters. In some respects, traditional pile design methods can still be applied to determine the ultimate loads that can be supported [14]. Due to these issues, new design methods are required to accurately calculate settlement and lateral deformation in early work [28,29] and updated conventional design approaches to account for the composite piles' unique properties. Given the data in Table 3, it is not surprising that they received the most citations because they incorporated this significant addition into the composite pile design. The type of composite pile represented in [28] is the one that appears the most frequently in keywords. The structural and geotechnical design of confined concrete FRP piles is considered when determining how to improve the design of composite piles.

4.2.1. Axial Loads

If the soil fails below the pile bottoms, around the pile-soil contact, or if the pile shafts are crushed by compression, FPR composite piles can fail under compression loads. Therefore, the load-bearing capacity of the FRP pile is determined by the value that is less: compressive strength and durability of the shafts or maximum load that causes soil failure and pile-soil contact. Compression tests provide a straightforward method for calculating the FRP pile compressive strength [28]. Naser, Hawwileh, and Abdalla [30] concluded that FRP-restricted concrete had a greater load capacity than plain concrete. A principle commonly acknowledged is that a pile's ultimate compression load capacity equals the sum of the load on the pile's end bearing and the friction on the pile's sides. The pile end-bearing load is based on the soil condition and the pile foundation's size, shape, and depth. It is not based on the pile material.

In contrast, the shear strength of the pile interface between the pile material and the soil depends on the pile material's roughness and the soil's condition. A designated design parameter must be used for the skin friction between the FRP composite piles and the soil to determine the ultimate bearing capacity. Three different approaches exist for calculating pile skin friction: the (α method) and (λ method) for fine-grain soil and the (β method) for granular soil. Commonly, skin friction, τs, values for piles in clay or sand are calculated using the (β method), which is represented below in Equations (1)–(3):

$$\tau_s = \sigma'_r \, \tan \delta_p \tag{1}$$

$$\tau_s = k \, \sigma'_v \, \tan \delta_p \tag{2}$$

$$\tau_s = \beta \, \sigma'_v \tag{3}$$

where $\sigma'_r$ = After installation, the horizontal functional tension acting around the pile shaft
$\sigma'_v$ = deep vertical functional stress.
K = coefficient of pressure in the horizontal plane.
$\delta_p$ = highest achievable pile-soil contact friction.

The FRP piles' frictional performance was tested in tank-stored clayey soil samples. The compression bearing capacity of FRP piles was revealed to be 5–40% greater than that of steel piles of the same size when the load transfer mechanisms of each were evaluated [25]. Giraldo Valez and Rayhani [31] conducted a comprehensive investigation analyzing the friction properties of FRP piles when placed against the clay. The results matched the conclusion of [25].

In addition, researchers in sand soil [31,32] conducted experimental investigations on the shear strength of the interface between FRP and sand. Frost and Han [33] discovered that the shear strength of the interface between FRP and sand depended on the normal stress, the relative roughness, and the angularity coefficient of soil. Regarding coefficient of interaction (CI), this relationship (Equation (4)) has been widely adopted to evaluate the interface shear strength's efficacy relative to the soil's internal shear strength.

$$CI = \tan \delta p / \tan \varnothing p \tag{4}$$

where $\delta p$ = maximal surface friction at the pile material's contact with the soil, and $\varnothing p$ is the soil angle with the greatest maximum internal friction. Figure 9 shows the coefficients of interaction (CI) against the roughness index (Ri). The roughness index is calculated by taking the greatest roughness of the surface of the pile material and dividing it by the mean grain size of the soil particles, also known as $D_{50}$. According to Figure 9, the interaction coefficients for the smooth FRP material range from 0.4 to 0.5, whereas the interaction coefficients for the rough FRP material range from 0.5 to 0.9.

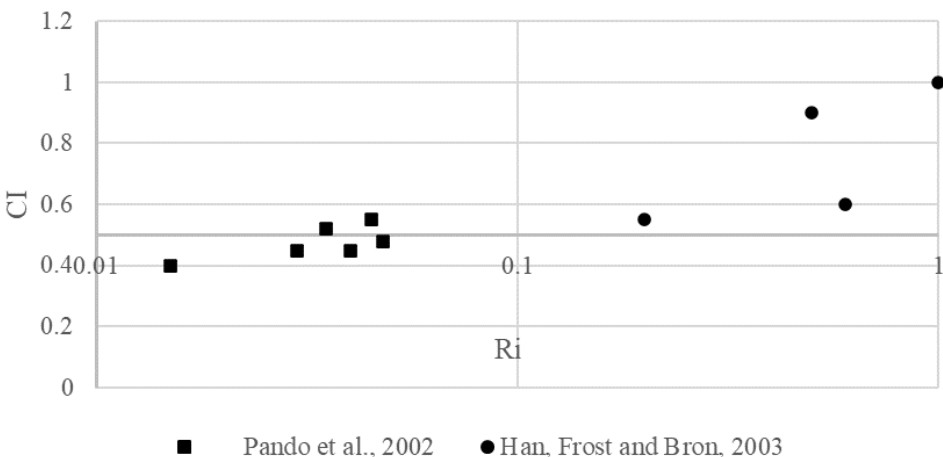

**Figure 9.** Coefficient of interaction (CI) between the FRP surface and the roughness index (Ri).

4.2.2. Lateral Loads

The maximum allowed deflection usually controls how a pile will act when subjected to lateral loads [34]. Pile bends are affected by the horizontal load, the soil's resistance to lateral movement, and the pile's bending stiffness, which is found by multiplying the second moment of the area by Young's modulus. The sub-grade reaction technique, the P-y method, the elastic continuum theory, and FE-based approaches are standard methods that may be used to analyze conventional piles when subjected to lateral stresses. Even though these approaches describe the pile as an elastic beam, this model may no longer be viable for composite pile types due to the relatively low shear modulus, which results in increased shear deformation [35].

Ma et al. [36] researched the composite pile's bearing properties and horizontal load transmission mechanisms and revealed that the pile foundation's lateral force against pile head displacement curves displays a steep decrease in section (typical piercing damage). The pile heads to one-third of the depth where stress and strain are concentrated. Consequently, they claimed that the lateral bearing capacity primarily depends on the soil's strength at contact with the pile or the displacement of the pile head.

Han, Frost, and Brown (2003) demonstrated that the following formula might be used to compute the normalized lateral displacement of FRP composite piles. This was done while taking into consideration FRP composite piles.

$$\text{w} = \frac{H\,w^-}{\sqrt[5]{\mathfrak{y}^3\left(I_{yy}E_{zz}\right)^2}} \tag{5}$$

where w = horizontal displacement (mm), $H$ = lateral force(kN), $w^-$ = normalised lateral displacement (mm), $E_{zz}$ = modulus of the longitudinal section in the *z*-axis of a pile (MPa), $I_{yy}$ = rotational inertia of a cross-section around the *y*-axis (mm$^4$), and $\mathfrak{y}$ = reaction constant for soil. Using a composite material with a higher shear modulus reduces horizontal displacement.

The shear modulus is a product derived from the section properties, such as the thickness and radius of the tube. By enhancing these qualities, the horizontal displacement can be limited.

Mirmiran, et al. [37] studied two types of concrete-filled FRP piles under axial and flexural stress. According to the investigation, over-reinforced specimens performed significantly better as beam columns and suffered 5–50 percentage points less displacement than their reinforced counterparts. According to the findings of Pando [14], shear deformations need to be considered in calculating subsequent deflections if there is a rise in the section's pile modulus ratio (E/G). Zyka and Mohajerani [38] highlighted a rising demand for more precise design approaches regarding confined concrete FRP piles' lateral and flexural load capability.

### 4.2.3. Earthquake Seismic Loads

Composite piles are designed to enhance the strength and stiffness of concrete and the ductility and corrosion resistance of surrounding tubes, making them suitable for use in seismic areas where soil and foundation conditions can be unpredictable. When subjected to seismic loads, composite piles can provide improved performance compared to traditional piles due to their unique combination of material properties. Several factors influence the behavior of composite piles under seismic loads, including the pile's design, the type of soil in which the pile is installed, and the intensity and duration of the seismic loads. Hosseini and Rayhani [39] evaluated the seismic performance of hollow fiber-reinforced polymer piles in liquefiable sand deposits using shaking table tests. Four glass fiber-reinforced polymer piles, four carbon fiber-reinforced polymer piles, and four aluminum piles were tested in soil-foundation models. The tests were conducted using earthquake ground motions. Results showed that the hollow fiber-reinforced polymer piles performed better than the aluminum piles, and the glass fiber-reinforced polymer piles performed better than the carbon fiber-reinforced polymer piles. The study suggests that hollow fiber-reinforced polymer piles can be a suitable alternative in seismically prone areas due to their favorable material characteristics. This study has been emphasized again by [40]; the authors report on two shaking table tests that studied the seismic performance of fiber-reinforced polymer (FRP) pile groups in saturated sand. The tests used a laminar shear box and different types of composite pile models (CFRP, GFRP, and aluminum). The piles were subjected to earthquake simulations from the 2010 Val-des-Bois and 1995 Kobe earthquakes. The results showed higher levels of excitation in the foundation compared to the soil, and the pile material affected the seismic response of the pile caps and superstructure. Aluminum piles had the highest acceleration response, while GFRP piles provided slightly better performance and could be a better alternative for traditional frictional piles in liquefiable soils. Abouelmaty, Elmasry, and Abdelaziz [41] studied the behavior of FRP composite piles under earthquakes, which introduced a dynamic model for a piles-cap system in a nearshore bridge. The results showed that although FRP piles are durable in harsh environments, their low modulus of elasticity reduces lateral stiffness and affects the

foundation system's efficiency during lateral earthquake excitation. This behavior should be considered when designing bridges in active seismic zones.

Furthermore, despite the earthquake load, the pile foundation still benefits from lateral support provided by the neighboring sand-tire mixture [42,43], thus safeguarding it against buckling and fracturing. This behavior of the pile foundation is distinct from that of a foundation supported solely by pure sand, which offers no lateral support during an earthquake. Nonetheless, an optimal length exists for reinforcing the sand-tire mixture beyond which its effectiveness becomes negligible.

It is discovered from this review that the limited availability of studies on composite piles subjected to earthquake seismic loads has implications for the understanding of their behavior and performance under these conditions. As a result, there is a need for further research in this area to establish more comprehensive and reliable data on the behavior of composite piles under earthquake seismic loads. This information would be useful for designing and constructing structures in seismic regions and contribute to improving the seismic performance of structures supported by composite piles.

## 5. Conclusions

The primary benefits of composite piles over conventional pilings are their low life-cycle costs and reduced environmental impact. Review articles are limited in these major shifts in foundation types and may be one-sided and subjective. This paper represents the first scientometric analysis of composite piles. After filtering, 252 articles were selected from Scopus. Keywords, countries with the highest citations, cutting-edge research institutes, and most-cited authors in composite piles were all investigated. The research articles from Scopus were used to create keyword clusters that helped identify historically divided types of composite piles and structural and geotechnical design requirements for composite piles (FRP piles).

Regarding composite piles, limited options offer extended service life and require less maintenance. However, certain types, such as SRP piles, plastic lumber piles, and fiberglass pultruded piles, could have a potential drawback in terms of structural performance due to their low-stiffness material. To overcome this issue, researchers can explore new composite piles that share service life and maintenance requirements while having better structural performance. Additionally, this research highlights the need for standardized testing and design principles for composite piles, as current pile design procedures are not comprehensive. Composite piles have a low stiffness due to the reduced modulus of compound materials, and their vertical and lateral load-settlement responses require different design approaches. Furthermore, it is essential to understand how earthquake, seismic, and lateral loads affect composite piles' short-term and long-term deflection.

For future prospects, and to ensure a bright future for FRP piles, it is important to conduct further investigations and simulations to evaluate their performance in various soil types, including cohesive and non-cohesive soils. Furthermore, studying the impact of combined loading conditions, especially in marine environments, on the performance of the FRP piles under lateral and axial loads is crucial. Researchers should also explore innovative composite pile designs with optimal structural behavior and comparable service life and maintenance requirements. Finally, comprehensive research on composite pile groups is necessary to gain insights into their behavior under axial and lateral loads and their arrangement within a group.

**Author Contributions:** Conceptualisation, F.A.-D., M.S., T.M.Č., A.G. and Z.Y.; Methodology F.A.-D., M.S., T.M.Č.; software, F.A.-D. and T.M.Č.; validation, T.M.Č., A.G. and Z.Y.; formal analysis, T.M.Č., M.S. and A.G.; investigation, M.S. and T.M.Č. original draft preparation, K.H., F.A.-D., M.S., T.M.Č., A.G. and Z.Y.; writing—review and editing, K.H., T.M.Č. and M.S.; supervision M.S., T.M.Č., A.G. and Z.Y. All authors have read and agreed to the published version of the manuscript.

**Funding:** This research received no external funding.

**Institutional Review Board Statement:** Not applicable.

**Informed Consent Statement:** Not applicable.

**Data Availability Statement:** The data presented in this study are included in the article. Further inquiries can be directed to the corresponding author.

**Conflicts of Interest:** The authors declare no conflict of interest.

**Sample Availability:** Not applicable.

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
