# Peer review of "A Systematic Review of the Geotechnical and Structural Behaviors of Fiber-Reinforced Polymer Composite Piles"

_geosciences, doi:10.3390/geosciences13030078_

Round 1

Reviewer 1 Report

In this manuscript, a systematic review on composite piles is investigated based on the data obtained from Scopus. VSOviewer software was used to spot the scientometrics based on mapping of knowledge. Data mining are conducted for the application of the composite piles. This paper is properly organized and the scientometric maps are clear. I recommend that this manuscript can be accepted after minor changes.

Detailed comments:

1.      Bibliometric subject word retrieval is the first step of mapping knowledge domain. Only "composite piles" as the keyword is retrieved in this paper, which may not be comprehensive. In most cases, researchers do not use the term composite piles. Some researchers may define their own term as pipe pile or FPR pipe pile, and so on. Therefore, it is suggested that some synonym of “composite piles” should be added in document retrieval to avoid missing any important references. After this, more irrelevant documents may be retrieved, but such documents can be excluded in the next manual screening step.

2.      Some punctuation marks and hyphens in the text are incorrect, such as line 76, line 186, line 242 etc.

3.      Some incorrect abbreviations appear in section 4.1, such as FL in line 196, SCR in line 206.

4.      Line 207: This is an incomplete sentence.

5.      Line 243: “pile’ ” should be “pile’s” or “piles’”.

6.      Line 272: “K” should be in lowercase.

7.      Line 287: Initial should be capitalized.

Author Response

Responses to reviewer 1

Dear reviewer, many thanks for your valuable feedback. We have addressed all mentioned comments as follows:

Seq.

Reviewer Comment

Author’s response

1.        

Bibliometric subject word retrieval is the first step of mapping knowledge domain. Only "composite piles" as the keyword is retrieved in this paper, which may not be comprehensive. In most cases, researchers do not use the term composite piles. Some researchers may define their own term as pipe pile or FPR pipe pile, and so on. Therefore, it is suggested that some synonym of “composite piles” should be added in document retrieval to avoid missing any important references. After this, more irrelevant documents may be retrieved, but such documents can be excluded in the next manual screening step.

Many thanks for this valuable point. In the revised paper, we have changed the title to (A Systematic Review of the Geotechnical and Structural Behaviors of Fibre Reinforced Polymer Composite Piles) to focus on one type of composite piles, which is the Fibre Reinforced Polymer Composite Piles. Additionally, we gave a general idea about other types of composite piles to avoid any confusion, which are:

·         Steel Core Plastic piles (SCP)

·         Structurally Reinforced Plastic piles (SRP)

·         Plastic Lumber piles (PL)

·         Fibreglass Pultruded piles (FP)

2.        

Some punctuation marks and hyphens in the text are incorrect, such as line 76, line 186, line 242 etc.

Corrected

3.        

Some incorrect abbreviations appear in section 4.1, such as FL in line 196, SCR in line 206.

Corrected as follows:

FL: This is a printing mistake; the correct abbreviation is (PL) from the Plastic Lumber piles.

SCR: the correct abbreviation is SCP.

Both errors have been corrected accordingly.

4.        

Line 207: This is an incomplete sentence.

Corrected

5.        

Line 243: “pile’ ” should be “pile’s” or “piles’”.

Corrected

6.        

 Line 272: “K” should be in lowercase.

Corrected

7.        

Line 287: Initial should be capitalized.

Corrected

Reviewer 2 Report

The paper consists only review of different papers about composite piles. However, it also should be mentioned, what is a purpose of gathering this information and if authors are planning their own research in field of composite piles. The plans of future research should be mentioned

The English language of the paper should be improved, for example:

- L .11 the phrase "Composite piles" is used twice in one line. Please rephrase it
-The paragraphs 4.1.2,  4.1.3, 4.1.4., 4.1.5. should not start as continuation of paragraph title, but the should start from the beginning
-etc.

Author Response

Response to reviewer 2

Dear reviewer, many thanks for your valuable feedback. We have addressed all mentioned comments as follows:

Seq.

Reviewer Comment

Author’s response

1.        

The plans of future research should be mentioned

The following paragraph has been added to the conclusion:

For future prospects, to ensure a bright future for FRPs, it's important to conduct further investigations and simulations to evaluate their performance in various soil types, including cohesive and non-cohesive soils. Furthermore, studying the impact of combined loading conditions, especially in marine environments, on FRP performance under lateral and axial loads is crucial. Researchers should also explore innovative composite pile designs that offer not only optimal structural behaviour but also have comparable service life and maintenance requirements. Finally, comprehensive research on composite pile groups is necessary to gain insights into their behaviour under axial and lateral loads, as well as their arrangement within a group.

2.        

The English language of the paper should be improved

The English language has been proofread by a native editor.

3.        

L .11 the phrase "Composite piles" is used twice in one line. Please rephrase it –

The paragraphs 4.1.2, 4.1.3, 4.1.4., 4.1.5. should not start as continuation of paragraph title, but the should start from the beginning

-etc.

Corrected by removing the repeated sentence.

Corrected accordingly.

Reviewer 3 Report

1.       The quality of presentation of the paper should be improved:  figures, missing references, especially and language of the text

2.       How did you get the information for Fig.2~4? How did the figures related to the content of the study.

3.       For Section 4, these papers might be helpful to strengthen the content: 1) Preliminary Study on the Behaviour of Fibre-Reinforced Polymer Piles in Sandy Soils; 2) FEM analysis of wasted tire chip and sand as construction material for piles; 3) A Workability Characterization of Innovative Rubber Concrete as a Grouting Material

4.       The references should be double checked, as some are missing in the text.

5.       For Fig.1, where did you get these picture. Citation is needed if not your own photo

Author Response

Response to reviewer 3

Dear reviewer, many thanks for your valuable feedback. We have addressed all mentioned comments as follows:

Seq.

Reviewer Comment

Author’s response

1.        

The quality of presentation of the paper should be improved:  figures, missing references, especially and language of the text

The quality of the presentation has been improved. Also, the missed references have been inserted in the required places, such as figures.

The English language has been proofread by a native editor.

2.        

How did you get the information for Fig.2~4? How did the figures related to the content of the study.

These figures were developed based on the Scopus database. In the revised version, this point was clarified.

These figures are important because they give a general idea about the history of the idea development (number of publications per year over the last three decades). Also, these figures give an idea about the types of publications and where published. Therefore, these figures are a guideline for the review process.

3.        

For Section 4, these papers might be helpful to strengthen the content:

1) Preliminary Study on the Behaviour of Fibre-Reinforced Polymer Piles in Sandy Soils;

2) FEM analysis of wasted tire chip and sand as construction material for piles;

3) A Workability Characterization of Innovative Rubber Concrete as a Grouting Material

Yes, these papers are very helpful and were included in the manuscript.

4.        

The references should be double checked, as some are missing in the text.

All references have been corrected using Endnote.

5.        

For Fig.1, where did you get these picture. Citation is needed if not your own photo

These photos were taken by the author in the city of Liverpool, UK.

Round 2

Reviewer 2 Report

OK, accepted

Reviewer 3 Report

The reviewer is happy with the amendement that authors have made. The manuscript can be now accepted .